

**Sulphur-rich volcanic eruptions triggered extreme hydrological events in**
**Europe since AD 1850**
**Cristina Di Salvo[1] Gianluca Sottili[1#]**
[1] Istituto di Geologia Ambientale e Geoingegneria IGAG-CNR, Rome [Italy].
*#Correspondence to: gianluca.sottili@uniroma1.it*
Keywords: sulphate aerosols, volcanic forcing, European climate, hydrological cycle





**Abstract**. Volcanic and anthropogenic aerosols, by reflecting solar radiation and acting as cloud condensation nuclei, play a key role in the global climate system. Given the contrasting microphysical and radiative effects of $SO_2$ on rainfall amounts and intensities, the combined effects of these two factors are still poorly understood. Here, we show how concentrations of volcanic sulphate aerosols in the atmosphere, as derived from Greenland ice core records, are strictly correlated with dramatic variations of hydrological cycle in Europe. Specifically, since the second half of the 19th century, the intensity of extreme precipitations in Western Europe, and associated river flood events, changed significantly during the 12-24 months following sulphur-rich eruptions. During the same period, volcanic $SO_2$ exerts divergent effects in central and Northern Europe, where river flow regimes are affected, in turn, by the substantial reduction of rainfall intensity and earlier occurrences of ice break-up events. We found that the high sensitivity of North Atlantic Sea Surface Temperature (SST) and North Atlantic Oscillation (NAO) to atmospheric $SO_2$ concentrations reveals a complex mechanism of interaction between sulphur-rich eruptions and heat exchange between Ocean and atmosphere with substantial impacts on hydrological regime in Europe.





## 1. Introduction

Fundamental for life, precipitation also plays a fundamental role in the redistribution of energy in the atmosphere as ~37% of the solar energy influx into the Earth's atmosphere is involved in the evaporation–condensation–freezing cycle. Highly variable in space, time and intensity, the emission of aerosols by volcanic eruptions may result into dramatic changes in precipitation patterns with large disagreement among models. It is widely accepted that global precipitation decreases one to two years after large explosive volcanic eruptions (Broccoli et al., 2003; Trenberth and Dai, 2007; Gu et al., 2007; Schneider et al., 2009; Gu and Adler, 2011; Iles et al., 2013). The decrease in global mean precipitation by volcanic aerosols is explained by the stabilization of the atmosphere due to the reduction of short-wave radiation reaching the surface thus resulting in a reduction of the evaporation (Bala et al., 2008; Cao et al., 2012). In addition, global circulation changes induced by sulphur-rich eruptions may also result into complex precipitations variations on a regional scale (e.g. in monsoon regions), with seasonal precipitation changes not yet well constrained by climate models. (Gu and Adler, 2011; Joseph and Zeng, 2011; Cao et al., 2012). Concerning the dynamics of volcanic forcing on seasonal and regional precipitation patterns, aerosols may produce slow or fast effects on hydrological cycle depending on whether ocean-atmosphere dynamics are involved or not (Rosenfeld et al., 2008). The fast effect of aerosol forcing, mostly related to solar radiation and cloud physics, has been investigated in more detail (Iles et al., 2013). On the other hand, the physical mechanisms of ocean-atmosphere interaction driving the slow effects (i.e., changes in seasonal and/or regional distribution of precipitation) are still poorly understood (Joseph and Zeng, 2011). In this perspective, evaluating the intensity of the effects of volcanic $SO_2$ conce.ntrations on hydrological cycle in different Europe climate zones may provide the needed evidence to support modelling work.

Here, we examine rainfall and river flow regimes since the second half of the 19[th] century in different European climate zones, as they relate to variations in volcanic $SO_2$ concentrations in the atmosphere. Specifically, we analyse short-term changes of river flows and rainfall regime in order to evaluate quantitatively the effects of sulphate aerosols on extreme precipitation, streamflow events and on hydrological cycle dynamics in Europe.



Precipitation and river flow data sets are analysed separately in four main climate zones
(Schneider et al., 2013) i.e., Mediterranean (MED), Temperate continental (TEMC),
Temperate transition (TEMT) and Temperate oceanic (TEMO) zones (Fig.1). The trigger
mechanism for extreme hydrological events by sulphur-rich eruptions may involve the
radiative forcing of sulphate aerosol over North Atlantic, as evidenced by anomalies in
North Atlantic sea surface temperature (SST) and in North Atlantic Oscillation values
recorded after sulphur-rich eruptions since 1850 AD.

**2. Dataset and analysis**
The analysed sulphate aerosol record (1850-1985 period), with an annual resolution based
on layer counting, derives from Greenland ice core analysis (Meese et al., 1997) in the
context of the Greenland Ice Sheet Project 2, (GISP2; Fig.1).The sulphate dataset clearly
record the thirteen VEI≥5 eruptions occurred on a global scale over the same time period
(Zielinski, 1995). The clear signatures of Icelandic sulphur-rich eruptions are also well
documented in the GISP2 record; the 1970 Hekla VEl3 eruption produced the highest $SO_2$
peak (i.e., 88 ppb) over the whole time series.
Rivers daily discharge data are from BfG Global Runoff Data Center and from local
governmental Institutions (Table 1); daily rain data are from NOAA Global Historical
Climatology Network. Specifically, for the MED climate zone, we examine the Tiber River
and the Collegio Romano rain gauge, for the TEMC zone, the Nemunas River and the
Vilnius Rain gauge, for the TEMT zone, the Elbe River and the Kremsmuenster rain
gauge, for the TEMO zone, the Thames River and the Armagh rain gauge (Fig. 1, Table

79    1).

Although the impact of strong volcanic $SO_2$ emissions on rainfall intensities and flash
floods are still poorly known, the land precipitation responses, in terms of monthly or yearly
amounts, to volcanic aerosols are reported to be significant from 1 to 3 years (Bala et al.,
2008). In particular, the maximum land mean precipitation reduction after volcanic
eruptions, as revealed by latent heat flux anomaly, occurs about 12 months after the
maximum reduction in the shortwave flux (Church et al., 2005). On this basis, we average
rainfall intensities response across multiple sulphate peaks beyond a fixed threshold



through a superposed epoch analysis. Specifically, we relate the annual $SO_2$
concentrations to the intensities of the twenty-five ($ERE_{25}$) and ten ($ERE_{10}$) most intense
precipitation episodes, recorded during year +1, as expressed in millimetres per 48 hours.
Then, we average $ERE_{25}$ data within two classes of $SO_2$ concentrations, i.e., years with no
detectable $SO_2$ and years with $SO_2 \geq 20$ ppb. After that, the relationship between annual
$SO_2$ record and rainfall and streamflow records during year +1 is considered.
A Monte Carlo technique was applied to asses the significance of the changes in extreme
rainfall intensities as a function of atmospheric $SO_2$ concentrations and to filter possible
effects of multiyear trends. Missing values of rainfall records were also assigned as
missing values in the Monte Carlo simulations. The statistical significance of the rainfall
intensity changes, $ERE_{25}$ and $ERE_{10}$, after $SO_2$-rich eruptions was evaluated by replacing
observed rainfall records with data from randomly selected years through 10,000
iterations. Mean $ERE_{25}$ and $ERE_{10}$ values were calculated from all years following sulphur-
rich eruptions associated to $SO_2$ concentrations above 40 ppb in the GISP2 record (Fig.2).
Then, the obtained mean $ERE_{25}$ and $ERE_{10}$ values were compared with results from
Monte Carlo simulations (10,000 iterations). The *p* values associated to rainfall intensities
changes after sulphur-rich eruptions is defined as the probability that the observed pattern
of $ERE_{25}$ and $ERE_{10}$, within individual climatic zones may derive from a random sampling
of the rainfall historical record. Thus, *p* values provide a quantitative estimation of the
significance of the detected relationship between $SO_2$ concentrations on rainfall intensities.
For each climate zone, the number of years with high-$SO_2$ concentrations within each
record and results from statistical analysis are summarised in Table 2.
The analysis of river streamflows is based on daily flow datasets for the Tiber, Nemunas,
Elbe and Thames rivers to calculate extreme day-to-day river flow increases ($\Delta Q_{day}$; Table
1). Specifically, for each year, $\Delta Q_{day}$ is defined as the 90th percentile of day-to-day
streamflow changes. All the considered rivers are characterised by dam systems for the
mitigation of flooding episodes in urbanised areas with potential effects on the analysed
$\Delta Q_{day}$ values. Thus, for each river drainage basin, we evaluated quantitatively the effects
of dam on the streamflow analysis with a statistical approach: concerning the Tiber river,
since 1921, (i.e., the starting point of present MED river flow analysis; Table 1), a possible



change-point in the flood record at the Ripetta hydrometric gauge took place in 1965, thus
possibly affecting ~31% of the total duration of the record. In fact, the Corbara dam, the
most important artificial structure to protect Rome from floods, operates since 1965 with a
water reservoir of 0.17 km$^3$ of active storage and a catchment area of 6,070 km$^2$. The
Corbara dam, by delaying the arrival time of flood waves from the upper Tiber, prevents
the superposition of flood waves so that the resulting flood waves can be smoothed (Natale
and Savi, 2007; Villarini et al., 2011)
In the Nemunas river, ice break-up events, rather than rainfall intensities, are the most
important controlling factor the discharge rate peaks. Long-term trends (1812-2006)
indicate that in the nineteenth century, ice cover remain unbroken on average for 30 days
longer than in the twentieth century (Stonevičius et al., 2008). Moreover, the construction of
the Kaunas Hydro Power Plant in 1959, recognised as the to the most impacting dam on
the Nemunas ice processes, decreased ice duration between 5 and 15 days on average
(Stonevičius et al., 2008).
The dominant flood threat for the Thames River, under favourable atmospheric conditions,
derives from surge tides. A complex system of embankments and floodwalls defends
London from the tidal regime. In recent times, re-profiling of beds and improvements to the
efficiency of weirs resulted in fewer floods in the lower Thames (Bell et al., 2012). To note,
we analysed the flow record at Teddington, the principal gauging station on the River
Thames, located at the tidal limit. The progressive construction of dams and embankments
on Elbe river (both in Czech Republic and Germany) and its tributaries (Vltava and Saale
in the Thuringian Forest) over the last two centuries makes difficult the definition of specific
major changing points in the day-to-day peak discharge series. A further element of
uncertainties in the streamflow record is determined by outflows and inundations occurring
as consequence of dike breaches during floods; (e.g., during the May-June 2013 flood in
central Europe when diffuse dike breaches took place along the Elbe river). On the other
hand, the role of tidal ranges on Elbe streamflow strongly decreases in the upstream
direction with no effect at the Neu-Darchau station (Table 1), about 220 km from the Elbe
mouth. To note, the middle Elbe part, including the Neu-Darchau station, is considered as
a semi-natural river without any river-regulating dams (Haberlandt et al., 2001).



Given the possible presence of changing points daily river flow records, firstly we analysed
the $\Delta Q_{day}$ time series of Tiber, Nemunas, Thames and Elbe rivers through the Mann–
Whitney approach (see Additional Informations). From $p$ obtained in the Mann–Whitney
statistical analysis (see Methods), we detected two main change points; the first concerns
the Tiber river with a change point ($p\sim0.02$) of $\Delta Q_{day}$ time series in 1965. The second
change point ($p\sim0.04$) is detected in the Nemunas river $\Delta Q_{day}$ time series in 1959. No
changes points were detected in the Thames and Elbe streamflow records. When no
statistically significant change-points are evidenced, we performed the $\Delta Q_{day}$ analysis, as
related to sulphate aerosol atmospheric records, over the entire record. On the other hand,
for the Tiber and Nemunas records, we split the record into two sub-series (i.e, before and
after the change-point); then we performed the $\Delta Qday$ analysis of the sub-series
separately (table 3).
The statistical significance of the $\Delta Q_{day}$ *vs.* sulphate concentrations relationship was
derived by applying the Monte Carlo method; specifically, the statistical analysis is based
on 10,000 iterations, by randomly sampling a number of $SO_2$ concentration value from the
historical record per iteration as the number of years within each quintile class (i.e., 20% of
years of the entire record). For each randomised quintile class, the mean $SO_2$ value is
calculated (Fig.3). Then, we evaluated quantitatively the probability, $p$, that the difference
of $SO_2$ concentrations between the first and the fifth $\Delta Q_{day}$ quintile in the historical record
may derive from a random sampling of the $SO_2$ record. For the Tiber and Nemunas rivers,
we analysed separately the two subseries after the detected changing points in 1965 and
1959, respectively. The subseries analysis gives level of significance $p<0.05$ for the Tiber
River and $p<0.02$ for the Nemunas rivers, thus excluding statistically significant effects of
change points on the $\Delta Q_{day}$ vs. $SO_2$ concentrations relationship (results in Table 3).
The Monte Carlo method was applied by assigning the pertinent $SO_2$ concentration value
(i.e., mean of the annual concentration values recorded during the earlier year) to the first
(lowest $\Delta Q_{day}$ values) and to the fifth (highest $\Delta Q_{day}$ values) quintiles interval obtained from
the 10,000 iterations. The null hypothesis of no changes of $\Delta Q_{day}$ values as a function of
$SO_2$ concentrations is verified from the width of the $SO_2$ concentration ranges within
randomised quintile classes.





The radiative forcing of sulphate aerosol over North Atlantic after sulphur-rich eruptions
was evaluated by considering seasonal SST and NAO variations since 1850 as a function
of $SO_2$ concentrations in the GISP2 record. Multiyear NAO trends are filtered by
normalising, within each year, the January to December monthly values between 0 and 1.
The SST and NAO datasets are available at *www.esrl.noaa.gov* and
*www.cpc.ncep.noaa.gov*, respectively.

**3. Results**
Figure 2 shows the response of $ERE_{25}$ intensities to increasing $SO_2$ concentrations in the
different European climate zones. In the MED area, years with $SO_2 \geq 20$ ppb are
characterised by $ERE_{25}$ intensities higher by 13.5 mm on average (standard deviation of
the mean, $\sigma_m$, 0.8; $p < 0.03$) with respect to pristine atmosphere years. In the TEMO zone,
$SO_2$ polluted conditions are associated with an increase of $ERE_{25}$ intensities by 13.1 mm
on average ($\sigma_m = 2.6$; $p < 0.01$). By contrast, in the TEMC zone, the values of $ERE_{25}$
decrease by 11.9 mm on average ($\sigma_m = 2.7$; $p < 0.19$). This trend is similar to that recorded in
the TEMT zone, where $ERE_{25}$ decreases by 16.0 mm on average ($\sigma_m = 2.9$; $p < 0.13$). To
note, when considering the most intense ten precipitation episodes, $ERE_{10}$, the effects of
$SO_2$ concentrations appear even more pronounced; in fact, in the MED area $ERE_{10}$
intensities increase by 13.9 mm on average ($\sigma_m = 1.9$; $p < 0.13$), while in the European
temperate oceanic zone they increase by 21.9 mm ($\sigma_m = 5.3$; $p < 0.01$). A more pronounced
trend concerns the $ERE_{10}$ values both in the TEMC zone ($ERE_{10}$ = -23.8 mm; $\sigma_m = 3.9$;
$p < 0.16$) and in the TEMT zone ($ERE_{10}$ = -19.9 mm; $\sigma_m = 2.6$, $p < 0.16$). This general trend in
rainfall intensity anomalies is relatively more evident when considering the effects of single
large volcanic eruptions; for example one year after the VEI6 1883 Krakatoa eruption,
$ERE_{10}$ values in the TEMO zone was affected by a +58.6 mm ($\sigma_m = 9.2$) change, with
respect to pristine atmosphere years (Fig.2). By contrast, in the TEMC and TEMT zones,
$ERE_{10}$ intensities changed by -62.3 mm ($\sigma_m = 7.5$) and -95.9 mm ($\sigma_m = 13.8$), respectively.
Now, we consider an independent dataset to verify if the observed $SO_2$-induced rainfall
extreme anomalies may have also induced detectable short-term effects on European
rivers flow regime. We analysed the daily streamflow data since the second half of the





19th century into the four European climate zones (Fig.1). In Figure 3, the plot of trends
was conducted by averaging the $SO_2$ concentration values (i.e., mean of the annual
concentration values recorded during the year preceding $SO_2$ peaks within fixed
concentration thresholds) to each $\Delta Q_{day}$ quintile interval. Results show that in the MED,
European TEMO and TEMC zones, $\Delta Q_{day}$ values increased significantly for increasing
values of atmospheric $SO_2$. Specifically, in the TEMO zone, the increase of $SO_2$ annual
mean concentrations from 11.9±3.5 to 28.5±7.6 ppb is followed by a factor ~2.3 $\Delta Q_{day}$
increase. This trend is even more marked in the MED region, where an increase of $SO_2$ by
a factor ~2.4 is followed by a factor >4 enhancement of $\Delta Q_{day}$. Even in the TEMC zone the
response of flow regime to increasing $SO_2$ concentrations shows a similar trend, with an
increase of $\Delta Q_{day}$ by a factor ~4.6 following an increase of $SO_2$ by a factor ~4.3. By
contrast, in the TEMT zone, to an increase of $SO_2$ by a factor ~4 corresponds a net
decrease of $\Delta Q_{day}$ by a factor ~3. The statistical significance of the river flow changes,
$\Delta Q_{day}$ values, as a function of $SO_2$ concentrations is summarised in table 3.

**4. Discussion**
Overall, it appears that, the response of rainfall and streamflow intensities to atmospheric
$SO_2$ concentrations defines a composite yet coherent geographical pattern in Europe. In
fact, after sulphur-rich eruptions, both rainfall and flash-flood intensities increase
significantly in the MED and TEMO zones, whilst an opposite trend is observed in the
TEMT zone. The TEMC zone represents an interesting exception, with a clear discrepancy
between the decrease of rainfall intensities and the increase of extreme streamflow
episodes following intense $SO_2$ peak concentrations. We note that annual discharge rate
peaks of the Nemunas River are mostly controlled by ice break-up events rather than by
rainfall intensities (Yoo and D'Odorico, 2002). Thus, the inconsistency between rainfall
intensities and river flow regimes might be related to some effect of atmospheric $SO_2$
concentrations on ice break-up events. In this perspective, it is widely accepted that
premature ice break-up events are associated with relatively more rapid runoff, usually due
to a combination of rapid melt and heavy rain (Beltaos and Prowse, 2001). Interestingly, after
sulphur-rich eruptions associated to $SO_2$ concentration values higher than 40 ppb in the



GISP2 record (twelve events since 1850) we found significant warmer temperatures of the
atmosphere in late winter and early spring in the TEMC zone (Fig.4). This atmospheric
warming is associated to a shift of ice break-up to early dates (i.e., by ~10 days, on
average), as revealed by spring discharge rate peaks in the Nemunas hydrograph.
Notably, the timing of ice break-up in northern Europe has been related to large-scale
atmospheric circulation processes over North Atlantic, as also evidenced by its close
relationship with the NAO (Livingstone 1999; Yoo and D'Odorico, 2002)
This picture suggests that the influence of sulphur-rich eruptions on the timing of ice
breaks and, more in general, on extreme hydrological events in Europe, can be related to
continental scale phenomena rather than to local-scale effects of $SO_2$ on hydrological
cycle dynamics.

**5. Conclusions**
We found that, since 1850, high $SO_2$ atmospheric concentrations are followed, during
year +1, by significant delayed responses of both the North Atlantic SST and NAO index
(Fig.5). This finding suggests a radiative forcing effect of sulphur-rich eruptions, as we
found that the twelve most intense $SO_2$ peaks (i.e., >40 ppb) since 1850 AD are followed,
during the year +1, by a North Atlantic SST negative summer anomaly up to ~0.1 °C. This
anomaly is followed, within 2-3 months, by a negative NAO phase. In addition, a clear
NAO positive phase is observed in February-March of the year +1. Interestingly, the
magnitude of positive and negative NAO anomalies increases for increasing $SO_2$
concentrations (Fig.5).
Although low latitude eruptions are reported to weakly enhance the NAO with relatively
warmer winter in the northern hemisphere (Robock and Mao, 1992; 1995; Stenchikov et al.,
2002; Hegerl et al., 2011) the response of NAO to sulphur-rich eruptions is not clearly solved
by climate models (Driscoll et al., 2012; Charlton-Perez et al., 2013). In this regard, the
Atlantic sea surface temperature (SST) is one of the most important governing factors for
the NAO and the atmosphere dynamics over most parts of the Northern Hemisphere
(Hurrell, 1995). Moreover, the lagged decrease of the NAO index following $SO_2$-induced
negative SST anomalies is coherent with the reported lagged covariability between



monthly SST and NAO (Czaja and Frankignoul, 2002; Wang et al., 2004). Negative NAO
phases corresponds to relatively weaker westerlies in the TEMC and TEMT zones with a
tendency toward blocking and greater frequency of meridional winds (Dettinger and Diaz,
2000; Wang et al., 2004). Under these blocked conditions, storms are steered toward
northern Europe or else directly into southern Europe; as a result, rainfall and streamflow
can be lowered over central Europe with negative NAO index. By consequence, in the
MED zone, negative NAO is associated to moist weather, as recorded by an increase in
river flow (Trigo et al., 2002; 2004). Regarding the TEMO zone, significant negative
correlations between NAO and regional rainfall amounts were observed in southern
England (Wilby et al., 1997) while positive correlations found in Scotland suggest a non
homogeneous geographical response of hydrological cycle to atmospheric circulation.
We propose a teleconnected mechanism for volcanically induced extreme hydrological
events in Europe. Specifically, the triggering mechanism of extreme rainfall and streamflow
events in Europe since 1850 after sulphur-rich eruptions can be explained by sulphate
aerosol radiative forcing over North Atlantic causing a net decrease of heat exchange
between Ocean and atmosphere through evaporation, precipitation and atmospheric-
heating processes. The results of this study display how sulphur-rich eruptions have
relevant significance in driving the frequency and intensity of rainfall and related floods in
Europe, with variable effects in different climate zones. Consequently, volcanic forcing of
hydrological cycle dynamics, superimposed to long term effects of the anthropogenic
climate change, needs to be addressed carefully in the context of densely populated
areas. As a consequence, this work can furnish a starting point for climate modelling
investigation, for reproducing past scenario and predictions at local scale and small
temporal resolution.

## Supplementary informations

Since the exact time of possible changing points on day-to-day peak discharge series of the
investigated rivers is unknown, we applied a non-parametric approach (Pettitt, 1979) for
determining the occurrence of a change point. This method allows the detection of significant
change in the mean of a time series. From the Mann–Whitney statistic $U_{t,N}$, we verified if two





samples $x_1, \ldots, x_t$ and $x_{(t+1)}, \ldots, x_N$ are from the same population. The test statistic $U_{t,N}$ is
given by:
$U_{t,N} = U_{t,-1,N} + \sum\limits_{j=1}^{N} \mathrm{sgn}(x_t - x_j)$ for $t = 2, \ldots, N$
The test determines the number of times a member of the first sample exceeds a member
of the second sample. The null hypothesis is the absence of one or more changing points.
The associated probabilities for the significance testing are given as:
$K_t = \mathrm{Max}_{1 \leq t \geq N} |U_{t,N}|$
and
$p \cong 2\exp[-6(K_t)^2 / (N^3 + N^2)]$
For p<0.05, a significant change point exists and represents the location of the division
point of the time series into two subseries.



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



**Figures**

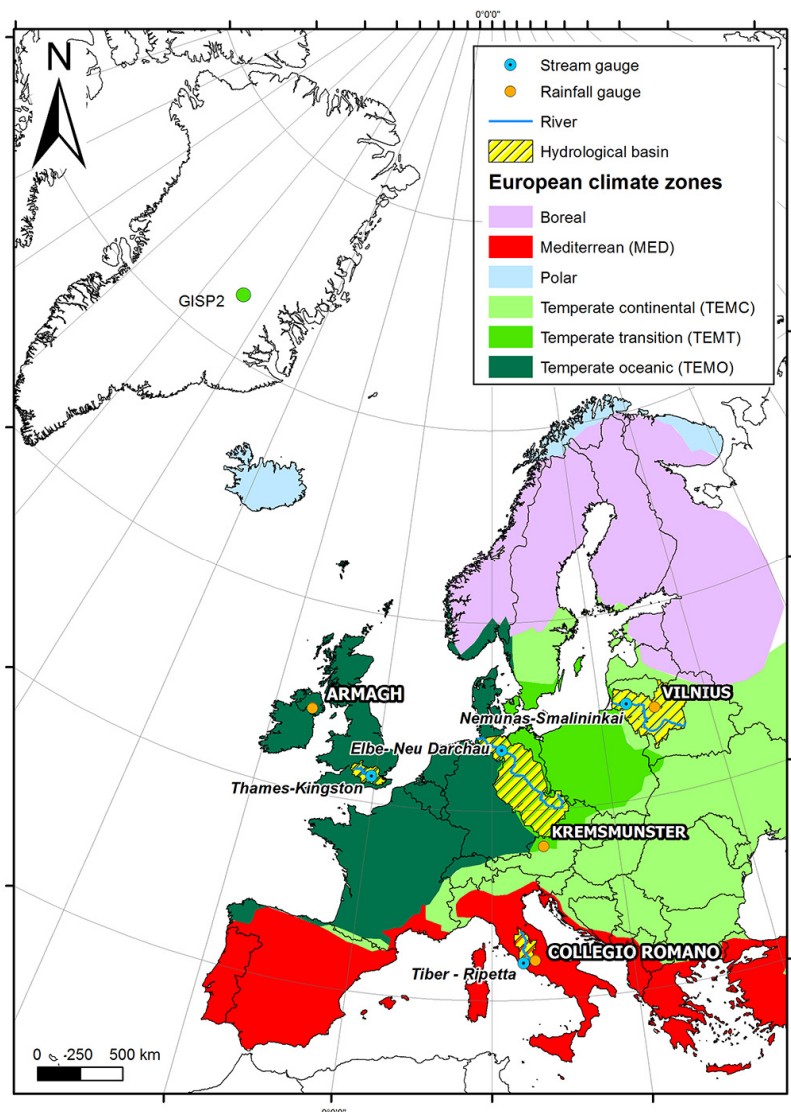



**Figure 1: Location of hydrometric and rainfall gauges considered in the present study**; the six European
climate zones (Meese et al., 1997) are also shown.



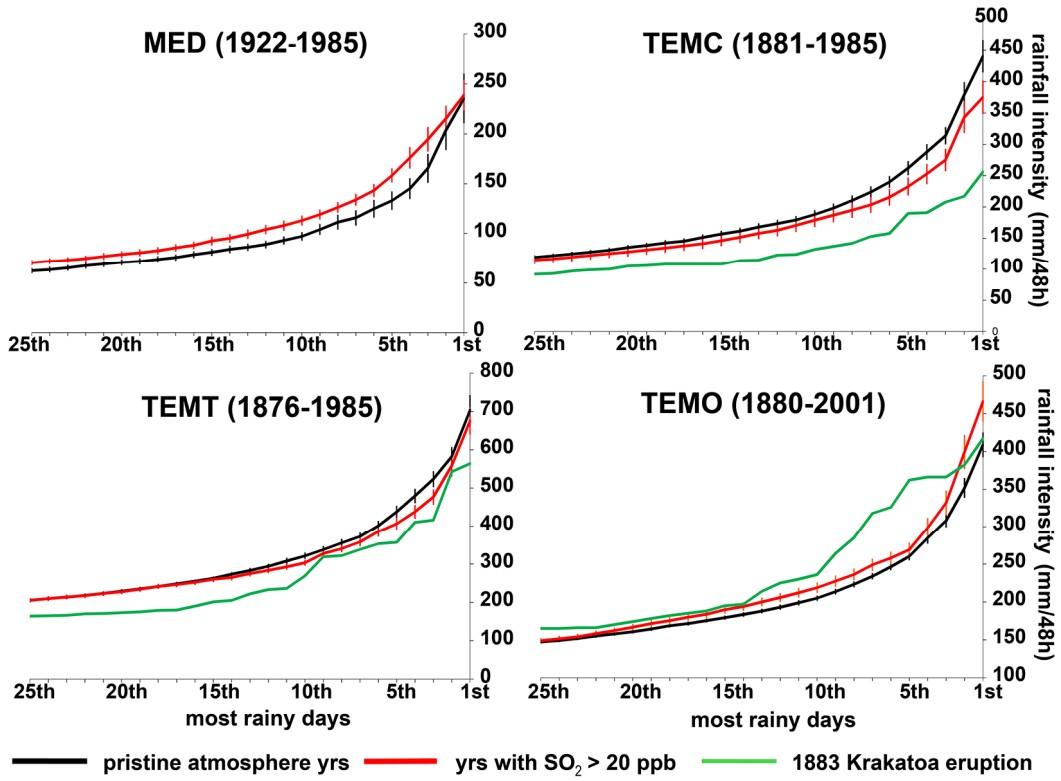



**Figure 2: Intensities of the 25 most rainy days (ERE₂₅) for years with no detectable volcanic SO₂ (black line) and years with SO₂≥20 ppb (red line).** Vertical bars are the standard deviations of the mean. Concentrations of volcanic aerosols above 20 ppb are associated to a significant increase of the ERE₂₅ intensities both in the MED area (mean value +10.7% at Collegio Romano rain gauge) and in the TEMO climatic zone (+1.4% at Armagh rain gauge) with respect to pristine atmosphere years. By contrast, in the TEMC (Vilnius rain gauge) and in the TEMT (Kremsmunster rain gauge) climatic zones, ERE₂₅ values decrease by 3.2 % and by 5.2 % on average, respectively. This general trend is more pronounced after large eruptions as, for example, after the 1883 VEI6 Krakatoa eruption (green line). SO₂ concentrations derive from GISP2 Greenland ice core record.( Schneider et al., 2013).



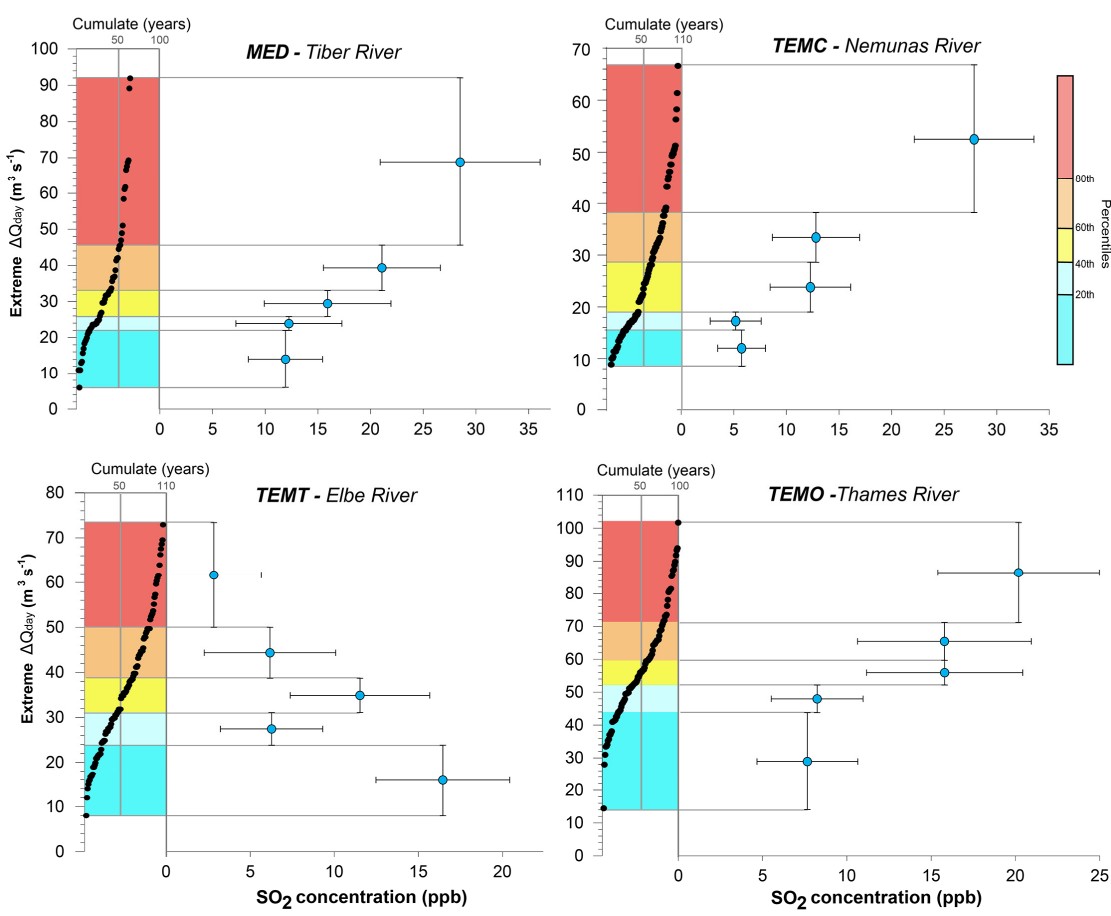

420

**Figure 3. Extreme day-to-day river flow increases (ΔQ$_{day}$) vs. SO$_2$ concentrations in Greenland ice core records (GISP2).** The Probability Distribution Function (PDF, black dots) of ΔQ$_{day}$ values was divided into five equal-sized groups (quintiles) and ordered from low to high ΔQ$_{day}$ values. SO$_2$ concentration values are the mean of the annual concentrations as determined within each ΔQ$_{day}$ quintile interval.





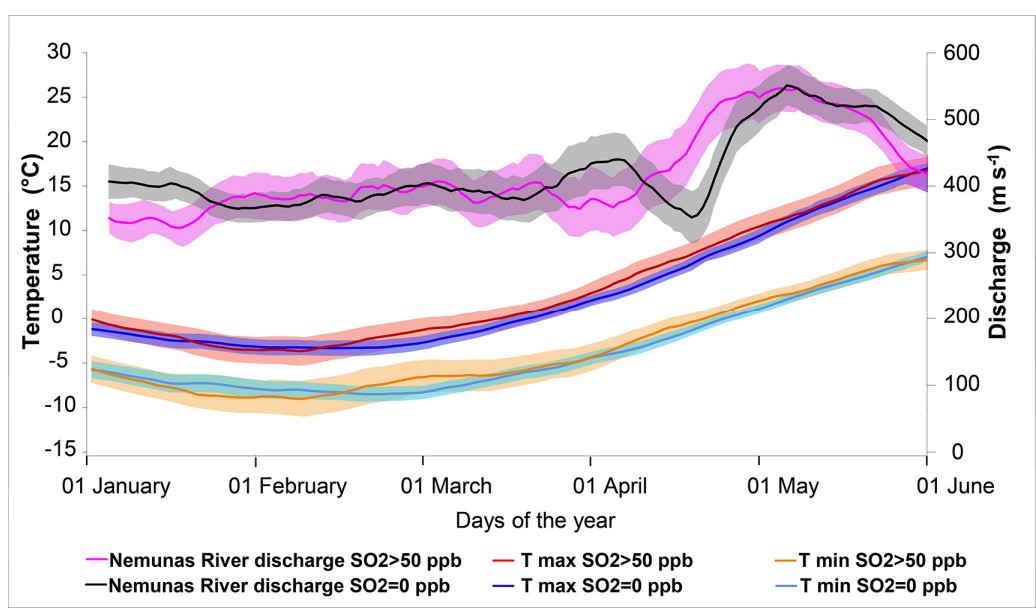

**Figure 4: Impact of volcanic SO₂ on the timing of ice breaks at the Nemunas River.** Nemunas River discharge at Smalininkai stream gauge (m³ s⁻¹) and maximum and minimum atmospheric temperatures at Vilnius (°C), for years with SO₂>50 ppb, years with SO₂=0 (pristine atmosphere). Both maximum and minimum temperatures show an increased trend during years with SO₂>50 ppb, with respect to pristine atmosphere years. The trend of discharge for years with SO₂>50 ppb clearly shows earlier dates for maximum flows, due to an earlier ice-break up, with respect to years with pristine atmosphere. Shaded areas are the standard deviation of the mean. Data source are reported in the text.



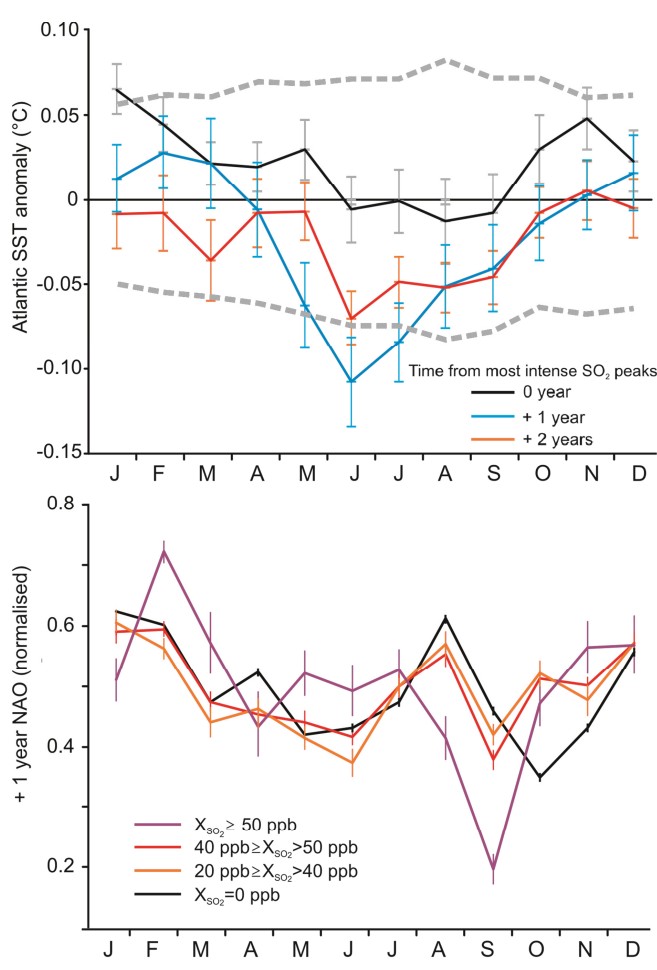

434

435

Figure 5: **Monthly impact of volcanic sulphate aerosols on Atlantic Sea Surface temperature (SST) and NAO index since 1850.** Post-volcanic Atlantic SST anomalies produced by the twelve most intense $SO_2$ peaks in the GISP2 record (i.e., $SO_2 \geq 40$ ppb compared with $SO_2 = 0$ ppb). Curves represent lag 0, lag +1 and lag +2 years, respectively, from sulphate peaks. Dashed lines denote the $1\sigma$ standard deviation from the monthly mean values over the entire record (upper). Sensitivity of NAO (normalised values) to increasing $SO_2$ concentrations values (lag +1 year) (lower). Vertical bars are the standard deviations of the mean.





**Table 1. Dataset, periods, hydrometric and rain gauge stations and references considered in**
**the river flow and rainfall analyses.**


| Climate zone | Period | Missing years | Station (*Country*) | Hydrometric (H) Rain gauge (R) | References |
|---|---|---|---|---|---|
| MED | 1922 - 1985 | 1928; 1934; 1937; 1941 | Collegio Romano (*Italy*) | R | * |
| | 1921 - 1985 | 1984 | Ripetta (*Italy*) | H (Tiber River) | ** |
| TEMC | 1881 - 1985 | 1915-17; 1943-44 | Vilnius (*Lithuania*) | R$^{\#}$ | *** |
| | 1877 - 1985 | 1930-32; 1943-45 | Smalininkai (*Lithuania*) | H (Nemunas River) | ** |
| TEMT | 1876 - 1985 | - | Kremsmuenster (*Austria*) | R | *** |
| | 1875 - 1985 | - | Neu-Darchau (*Germany*) | H (Elbe River) | ** |
| TEMO | 1880 - 1985 | - | Armagh (*United Kingdom*) | R | *** |
| | 1883 - 1985 | - | Kingston (*United Kingdom*) | H (Thames River) | ** |


*Notes:*
*\* Ufficio Idrografico e Mareografico Regione Lazio [UIRL], Centro Funzionale*
*(http://www.idrografico.roma.it/)*
*\*\* Global Runoff Data Centre [GRDC]. Koblenz, Federal Institute of Hydrology (BfG), (2014).*
*\*\*\* Global Historical Climatology Network [GHCN],NOAA Satellite and Information Service,*
*R$^{\#}$ Temperature analysed in figure 4 are from the Vilnius station.*



**Table 2. Statistical significance of the effects of SO₂ on rainfall intensities from Monte Carlo**
**method. Within individual climate zones, the number of years with high-SO₂ concentrations (≥40**
**ppb) corresponds to the number of randomly selected years within the GISP2 record for Monte**
**Carlo simulations ($10^4$ iterations).**

| Climate zone | Station (*Country*) | Record duration yrs (missing yrs) | yrs with SO₂ ≥40 ppb | p (ERE$_{25}$) | p (ERE$_{10}$) |
|---|---|---|---|---|---|
| MED | Collegio Romano (*Italy*) | 65 (*4*) | 9 | <0.03 | <0.13 |
| TEMC | Vilnius (*Lithuania*) | 106 (*5*) | 12 | <0.19 | <0.16 |
| TEMT | Elbe (*Germany*) | 111 (*0*) | 12 | <0.13 | <0.16 |
| TEMO | Thames (*UK*) | 107 (*0*) | 12 | <0.01 | <0.01 |


**Table 3. Results from Monte Carlo method for the statistical significance of the effects of SO₂ on**
**streamflow extreme events.**

| Climate zone | River (*Country*) | Record duration yrs (*missing yrs*) | Mean SO₂ concentration ($\sigma_m$) in ppb | | *p* |
|---|---|---|---|---|---|
| | | | ΔQday lowest (1st) quintile | ΔQday highest (5th) quintile | |
| MED | Tiber (*Italy*) | 65 (*1*) | 11.9 (*3.5*) | 28.5 (*7.6*) | <0.01 |
| TEMC | Nemunas (*Lithuania*) | 109 (*6*) | 6.1 (*2.3*) | 28.2 (*5.7*) | <0.001 |
| TEMT | Elbe (*Germany*) | 110 (*0*) | 7.6 (*3.0*) | 20.1 (*4.8*) | <0.01 |
| TEMO | Thames (*UK*) | 103 (*0*) | 18.2 (*4.0*) | 4.6 (*2.8*) | <0.02 |
