# Peer review of "Sulphur-rich volcanic eruptions triggered extreme hydrological events in 2 Europe since AD 1850"

_Climate of the Past, 2016_

## Referee Comment (RC1) · Anonymous Referee #1 · 20 Jun 2016

This paper presents an analysis of the relationship between years with elevated volcanic sulphate deposition in a Greenland ice core (GISP2), and riverflow or rain gauge data from four climate zones across Europe. The analysis is detailed, and suggests that there is evidence for a link between the volcanic forcing and the hydrological response in datasets since 1850 CE.

The major weakness of the paper is that the analysis deals almost entirely with derived datasets; and makes no attempt to show the reader the time-series of the data. There is also a lack of clarity about the precise nature of the primary datasets that are used - making it difficult for the reader to compare and contrast results across different studies (e.g. what happened in 1913? or 1982?). It is essential that, in revision, the authors at

least provide full citations and links to all of the datasets used in the analysis; and provide a fuller explanation of how they chose the datasets that are used in this analysis, and of the limitations (time, space, resolution) of each dataset.

Specific comments.

line 17 'since the second half..'- actually your records don't start until the 1877? The analysis stops at 1985.

Line 32 how can there be 'large disagreement among models' but yet be 'widely accepted' that global precipitation decreases? More careful and critical analysis of the published literature is needed here: what do the empirical observational data suggest? What do different models suggest? See for example the recent paper by Liu et al 'Global monsoon precipitation responses..' Scientific Reports, | 6:24331 | DOI: 10.1038/srep24331

Line 67 – explain why you chose to use the GISP2 record, and give the proper citation to the timeseries that you analyse in this paper (it isn't Meese et al., 1997, which presents an age model). Why not present this record for completeness; this would help the reader understand the nature of the analysis? Is it actually annual (rather than biannual?). How does the age/identification of volcanic events compare to those in Sigl et al (2015, Nature doi:10.1038/nature14565)? How are the sulphate concentrations you quote determined? (is it the sulphate deposition record, or an atmospheric aerosol record; how did the authors determine the volcanic contribution to the sulphate?) Why would you treat Icelandic (local) and non-local eruptions in the same way in your analysis? Does the time range of your analysis stop at 1985? why not extend this to include the last 30 years of analysis (and another VEI 6 eruption)?

Line 85-86 what exactly does your method do? What is the fixed threshold, and how did you identify it?

Line 237 '12 events..' please specify. Are these all large explosive eruptions? Is there

[Figure]

any dependence on hemisphere/latitude of the source eruption?

Figure 1 – caption has the wrong citation?

---

## Referee Comment (RC2) · Anonymous Referee #2 · 22 Jun 2016

The authors present a study on the effects of volcanic eruptions on extreme precipitation and river run-off in Europe over the last 160 years approximately. They apply a superposed epoch analysis to estimate the changes in the number of extreme precipitation events (2-day extremes) in the years following eruptions relative to normal years. They investigate four rain gauges and river discharge data of main rivers in Western Europe. The study finds a response of precipitation and run-off to volcanic eruptions in all four rain-gauges, with higher extreme precipitation in the mediterranean region and temperate oceanic western Europe and weaker extreme precipitation in continental Europe and in the transition zone. My impression of the manuscript is mixed. The first part containing the statistical analysis f the response to volcanic eruptions seems

to be broadly correct and the results here are interesting, although the magnitude of the signal does not seem to be very strong. There are a few minor points concerning this section that may require the attention of the authors. However, sections 4 and 5 seemed to me rather poor. My first concern is that section 4 (Discussion) still contains results that have not been included in section 3, and this is distracting. For instance, Figure 4 and Figure are first refered to in the Discussion section. These two figures, however, contain important information for the overall interpretation of the results. More importantly, the Discussion on the role of the North Atlantic Oscillation and North Atlantic sea-surface temperatures is rather shallow, with very speculative reasoning that is not supported by the results presented in this study nor by results of previous published studies. I explain in my detail my concerns regarding tis section below as well.

My overall recommendation is that this manuscript requires major revisions, and in the case of the last two sections these revisions should, in my opinion, quite substantial.

In the following list, the relevant sentence r paragraph in the manuscript is copied first, followed by my comment

1.

missing values in the Monte Carlo simulations. The statistical significance of the rainfall intensity changes, ERE25 and ERE10, after SO2-rich eruptions was evaluated by replacing observed rainfall records with data from randomly selected years through 10,000 iterations.

The Monte Carlo procedure is not totally clear to me. Were whole years resampled as a block, or were individual days resampled ? The sentence seems to indicate that entire years were resampled, but what is the rationale for this ? By resampling entire years, the values of the highest 25th events are not independent , so that the statistical significance is much more difficult to establish. I wonder if re-sampling individual days or 2-day blocks, from the same season would have been a better strategy.

different European climate zones. In the MED area, years with SO2$\geq$20 ppb are characterised by ERE25 intensities higher by 13.5 mm on average (standard deviation of the mean, $\sigma$m, 0.8; p<0.03) with respect to pristine atmosphere years. In the TEMO zone,

What is the meaning of 'standard deviation of the mean' ? If I understood properly, the mean is calculated by taken all years after volcanic eruptions. Is the standard deviation of the mean some type of bootstrap estimation or is it the sample standard deviation divided by n-1 ? The reader would appreciate a clearer language here. Also, what is the meaning of p here ? Is it the level of significance of the differences of the means between post-volcanic years and pristine years ? If yes, the language is unclear.

values of atmospheric SO2. Specifically, in the TEMO zone, the increase of SO2 annual mean concentrations from 11.9$\pm$3.5 to 28.5$\pm$7.6 ppb is followed by a factor $\sim$2.3 $\triangle$Qday increase. This trend is even more marked in the MED region, where an increase of SO2 by

'by a factor 2.3 $\triangle$Qday is unclear'. I think the authors man an increase by a factor 2.3

Overall, it appears that, the response of rainfall and streamflow intensities to atmospheric SO2 concentrations defines a composite yet coherent geographical pattern in Europe. In

I think this conclusion is too far-fetched. The study has analyzed four rain-gauges. Considering that rain is spatially quite variable, it is nut justified to extrapolate these results to whole four regions

5.

Conclusions We found that, since 1850, high SO2 atmospheric concentrations are followed, during year +1, by significant delayed responses of both the North Atlantic SST and NAO index (Fig.5). This finding suggests a radiative forcing effect of sulphur-rich eruptions, as we

Figure 5 is quite confusing. It is not really well described in the next, and it is distracting that it appears cited in the discussion section, as I mentioned earlier. I have several concerns: why is the standard deviation of the SSTs calculated over the whole period shown ? It seems to me that the authors want to show a statistically significance difference in the mean of the SSts in post-volcanic years and in pristine years. The standard deviation is not informative because it is the difference in the mean of two populations. The significance of these differences depends on the magnitude of the difference, the pooled standard deviations and the sample size. This third factor is not included in the figure. The results concerning the NAO (lower panel) do not seem to indicate a strong, or even statistically significance systematic response. The red and black lines are pretty close to each other and most of the time within the uncertainty ranges (= standard deviations of the mean?) Again, I think that this figure s not showing how significant the response of the NAO is, and nevertheless, even the sign of the response changes with time lag. For instance, it seems significant for August and September and then not significant in June, Juy and October. Can this be just a random effect ?. What are the physical mechanism by which the sign of the response may change ?

by climate models (Driscoll et al., 2012; Charlton-Perez et al., 2013). In this regard, the Atlantic sea surface temperature (SST) is one of the most important governing factors for the NAO and the atmosphere dynamics over most parts of the Northern Hemisphere (Hurrell, 1995). Moreover, the lagged decrease of the NAO index following SO2-induced

This paragraph is to me quite problematic. The question of whether SST anomalies are driving NAO variability and atmospheric circulation at midlatitudes in general is being discussed for at least 25 years, and there are studies with very opposing views. This section cites some of the studies by Czaja and Frankignoul and by Wang et al, that do indicate some response to SST anomalies, but there are may others that show no response. For instance, the paper by Sutton and Hodson (Science, 2005,doi:10.1126/science.1112666) indicates that ' However, thus far the evidence for an Atlantic link is mainly circumstantial, being derived from observations and showing correlation rather than causality' . This is a sign that, even in 2005, this question was far from settled, and actually it is not settled yet. Yet, the authors discuss this point rather superficially, as if it were widely accepted that NAO variability is driven by North Atlantic SSTs. Even the studies showing a response of the NAO to SSTs admit that the signal is weak. The paper by Sutton and Hodson identifies a response in the low-frequency band of the spectrum, i.e. not at interannual time scales, which would be the relevant time scales here, but at decadal timescales. Furthermore, the paper by Hurrel et al (1995) does not mention that SSTs are the most important driver of the NAO. Actually, if I understood that paper properly, it does not deal with the driving factors of the NAO. The statistical connections between the NAO and SSTs shown there are interpreted as a response of the SSTs to NAO forcing, which is the opposite interpretation given by the authors here.

7

We propose a teleconnected mechanism for volcanically induced extreme hydrological events in Europe. Specifically, the triggering mechanism of extreme rainfall and streamflow events in Europe since 1850 after sulphur-rich eruptions can be explained by sulphate aerosol radiative forcing over North Atlantic causing a net decrease of heat exchange between Ocean and atmosphere through evaporation, precipitation and atmospheric- heating processes. The results of this study display how sulphur-rich eruptions have relevant significance in driving the frequency and intensity of rainfall

and related floods in Europe, with variable effects in different climate zones.

To be honest, I do not think that this study has shown anything of this sort. It has not analysed radiative forcing, nor ocean-atmosphere heat exchange, nor evaporation, and it has analysed only four rain-gauges in Europe. I found the claims contained in this paragraph completely unsubstantiated.

---

## Author Comment (AC1) · 16 Aug 2016

Anonymous Referee #1 This paper presents an analysis of the relationship between years with elevated volcanic sulphate deposition in a Greenland ice core (GISP2), and riverflow or rain gauge data from four climate zones across Europe. The analysis is detailed, and suggests that there is evidence for a link between the volcanic forcing and the hydrological response in datasets since 1850 CE.

The major weakness of the paper is that the analysis deals almost entirely with derived datasets; and makes no attempt to show the reader the time-series of the data. There is also a lack of clarity about the precise nature of the primary datasets that are used - making it difficult for the reader to compare and contrast results across different studies

(e.g. what happened in 1913? or 1982?). It is essential that, in revision, the authors at least provide full citations and links to all of the datasets used in the analysis; and provide a fuller explanation of how they chose the datasets that are used in this analysis, and of the limitations (time, space, resolution) of each dataset.

Specific comments:

Comment by referee#1; Line 17 'since the second half..'- actually your records don't start until the 1877? The analysis stops at 1985.

Answer to comment: we agree and we will correct the title considering 1877 as starting year for our elaboration, instead of 1850.

Comment by referee#1; Line 32 how can there be 'large disagreement among models' but yet be 'widely accepted' that global precipitation decreases? More careful and critical analysis of the published literature is needed here: what do the empirical observational data suggest? What do different models suggest? See for example the recent paper by Liu et al 'Global monsoon precipitation responses..' Scientific Reports, | 6:24331 | DOI: 10.1038/srep24331

Answer to comment: Indeed, on the basis of the cited literature, we confirm that it is widely accepted that, on a global scale, annual precipitations decrease from one to two years after large explosive volcanic eruptions. However, on a regional scale, it is also accepted that there can be complex precipitations variations, as, for example, in monsoon regions (Wegmann et al., 2014; Liu et al., 2016); for this reason , we propose a rainfall intensity analysis on a local (i.e., individual basins) scale. However, in the revised version of the manuscript "large disagreement among models"could be deleted in order to better explain the above mentioned role of local trends on masking and/or filtering general, global scale patterns.

Comment by referee#1; Line 67 – explain why you chose to use the GISP2 record, and give the proper citation to the timeseries that you analyse in this paper (it isn't

Meese et al., 1997, which presents an age model). Why not present this record for completeness; this would help the reader understand the nature of the analysis? Is it actually annual (rather than biannual?). How does the age/identification of volcanic events compare to those in Siglet al (2015, Nature doi:10.1038/nature14565)? How are the sulphate concentrations you quote determined? (is it the sulphate deposition record, or an atmospheric aerosol record; how did the authors determine the volcanic contribution to the sulphate?) Why would you treat Icelandic (local) and non-local eruptions in the same way in your analysis? Does the time range of your analysis stop at 1985? why not extend this to include the last 30 years of analysis (and another VEI 6 eruption)?

Answer to comment: As also mentioned by the referee#1, the correct primary data source for GISP2 is: "Greenland Ice Sheet Project 2 Science Management Office (1993), GISP2 core data book, 114 pp., Univ. of N. H., Durham". For sake of clarity, we will add this reference when presenting the GISP2 dataset. We will also explain that GISP2 record provides a continuously dated record of annual SO2 accumulation (Meese et al., 1994), in which the depth-age scale was obtained with the use of specific, independent techniques to count annual layers in the core. In our analysis we consider the concentration of volcanic SO2 over the Norh Atlantic region, as derived from the GISP record, irrespectively of the SO2 source areas. In fact, in our analysis, the effects of SO2 on the European hydrological cycle are not related to the total amount of SO2 released by individual volcanic events nor to the intensity of volcanic eruptions; indeed, in our model, the effects of SO2 are strictly related to the concentration of SO2 in the North Atlantic region. Concerning the analysed time range, we confirm that our analysis stops at 1985, i.e. the temporal window reported in the GISP2 project.

Comment by referee#1; Line 85-86 what exactly does your method do? What is the fixed threshold, and how did you identify it?

Answer to comment: Since the intensity of SO2 peaks is neither related to the intensity

of volcanic eruptions nor their geographical locations, we based our analysis on SO2 concentration values. Specifically, the effects of annual SO2 concentrations on rainfall intensities (i.e., twenty-five (ERE25) and ten (ERE10) most intense precipitation episodes) was evaluated by considering two classes of SO2 concentrations, i.e., years with no detectable SO2 and years with SO2$\geq$20 ppb. We remark that the number of years with SO2$\geq$20 ppb represents at least the 10% of individual rain gauge records. We propose that, in the revised version of the table 2, we could add the number of years with SO2$\geq$20 ppb for each rain gauge record.

Comment by referee#1; Line 237 '12 events.' please specify. Are these all large explosive eruptions? Is there any dependence on hemisphere/latitude of the source eruption?

Answer to comment: Again, we remark that the intensity of SO2 peaks is not related to the intensity of volcanic eruptions nor their geographical locations; in the revised version of the manuscript, we could report more details on the eruptive styles associated to the most intense SO2 peaks.

Comment by referee#1; Figure 1 – caption has the wrong citation?

Answer to comment: We will correct the caption with the correct citations as follow: "Figure 1: Location of hydrometric and rainfall gauges considered in the present study; the six European climate zones based on information provided by EUCA15000 (Schneider et al., 2013) and the position of Greenland Ice Sheet Project 2 (GISP2) are also shown."

---

## Author Comment (AC2) · 16 Aug 2016

The authors present a study on the effects of volcanic eruptions on extreme precipitation and river run-off in Europe over the last 160 years approximately. They apply a superposed epoch analysis to estimate the changes in the number of extreme precipitation events (2-day extremes) in the years following eruptions relative to normal years. They investigate four rain gauges and river discharge data of main rivers in Western Europe. The study finds a response of precipitation and run-off to volcanic eruptions in all four rain-gauges, with higher extreme precipitation in the mediterranean region and temperate oceanic western Europe and weaker extreme precipitation in continental Europe and in the transition zone. My impression of the manuscript is mixed. The first part containing the statistical analysis of the response to volcanic eruptions seems to be broadly correct and the results here are interesting, although the magnitude of the signal does not seem to be very strong. There are a few minor points concerning this section that may require the attention of the authors. However, sections 4 and 5 seemed to me rather poor. My first concern is that section 4 (Discussion) still contains results that have not been included in section 3, and this is distracting. For instance, Figure 4 and Figure are first refered to in the Discussion section. These two figures, however, contain important information for the overall interpretation of the results. More importantly, the Discussion on the role of the North Atlantic Oscillation and North Atlantic sea-surface temperatures is rather shallow, with very speculative reasoning that is not supported by the results presented in this study nor by results of previous published studies. I explain in my detail my concerns regarding tis section below as well. My overall recommendation is that this manuscript requires major revisions, and in the case of the last two sections these revisions should, in my opinion, quite substantial. In the following list, the relevant sentence r paragraph in the manuscript is copied first, followed by my comment

1. Manuscript text: "missing values in the Monte Carlo simulations. The statistical significance of the rainfall intensity changes, ERE25 and ERE10, after SO2-rich eruptions was evaluated by replacing observed rainfall records with data from randomly selected years through 10,000 iterations."

Comment by referee#2: The Monte Carlo procedure is not totally clear to me. Were whole years resampled as a block, or were individual days resampled ? The sentence seems to indicate that entire years were resampled, but what is the rationale for this ? By resampling entire years, the values of the highest 25th events are not independent , so that the statistical significance is much more difficult to establish. I wonder if resampling individual days or 2-day blocks, from the same season would have been a better strategy.

Answer to comment: Our analysis evidenced significant differences in ERE10 and ERE25 intensities between years with high (i.e., $\geq$20 ppb) and years with no detectable SO2 concentrations. The rationale of the Monte Carlo procedure consists in establishing quantitatively the probability, p, that rainfall intensity anomalies observed during years with SO2$\geq$20 ppb may derive from a random sampling of the historical record. Specifically, first we compared: i) the differences between ERE10 and ERE25 values recorded after high-SO2 years and ii) ERE10 and ERE25 recorded after pristine atmosphere years from historical records. Then, we reshuffled the historical record like a deck of cards, thus obtaining 10,000 synthetic time series by reassigning ERE10 and ERE25 values to random years of occurrence. Then we repeat the analysis for the 10,000 synthetic time series, thus obtaining the synthetic distribution of the 10,000 differences between the intensities of ERE10 and ERE25 during years with high (i.e., $\geq$20 ppb) and years with no detectable SO2 concentrations. Unlike the historical record, the randomised time series show no significant (or very low) probability, p, dependence of ERE10 and ERE25 on SO2 concentrations (Table 2).

2. Manuscript text: different European climate zones. In the MED area, years with SO2 20 ppb are characterised by ERE25 intensities higher by 13.5 mm on average (standard deviation of the mean, m, 0.8; p<0.03) with respect to pristine atmosphere years. In the TEMO zone,

Comment by referee#2: What is the meaning of 'standard deviation of the mean' ? If I understood properly, the mean is calculated by taken all years after volcanic eruptions. Is the standard deviation of the mean some type of bootstrap estimation or is it the sample standard deviation divided by n-1 ? The reader would appreciate a clearer language here. Also, what is the meaning of p here ? Is it the level of significance of the differences of the means between post-volcanic years and pristine years ? If yes, the language is unclear.

Answer to comment: Yes, the mean is calculated by taking all years after volcanic eruptions. The standard deviation of the mean is defined as follow:

Where N is the number of days, i.e., ten and twentyfive for ERE10 and ERE25, respectively and ïĄş is the standard deviation of ERE10 and ERE25 values, as expressed in mm. As defined above, p is the probability that that rainfall intesity anomalies observed during years with SO2$\geq$20 ppb may derive from a random sampling of the historical record.

3. Manuscript text: values of atmospheric SO2. Specifically, in the TEMO zone, the increase of SO2 annual mean concentrations from 11.9 3.5 to 28.5 7.6 ppb is followed by a factor 2.3 Qday increase. This trend is even more marked in the MED region, where an increase of SO2 by

Comment by referee#2: 'by a factor 2.3 Qday is unclear'. I think the authors man an increase by a factor 2.3

Answer to comment: True, in the revised version of the manuscript we will modify this point accordingly.

4. Manuscript text: Overall, it appears that, the response of rainfall and streamflow intensities to atmospheric SO2 concentrations defines a composite yet coherent geographical pattern in Europe. In

Comment by referee#2: I think this conclusion is too far-fetched. The study has analyzed four rain-gauges. Considering that rain is spatially quite variable, it is nut justified to extrapolate these results to whole four regions

Answer to comment: We fully agree that a higher number of rain gauges could significantly improve the statistical significance of the analysis; actually we considered the longest available time series of rainfall and river discharge in Europe, corresponding to four of the largest hydrological basins in the considered European different climate zones. Our results and conclusions refer to the four investigated basins, and consider some general elements (i.e., intensity of rainfall episoded and associated extreme streamflow intensities) to investigate possible links between hydrological events

and climatic zone trends (as for earlier ice break –up in TEMC zone). We propose a smoother sentence to describe our general findings, as: "Overall, it appears that, the response of rainfall and streamflow intensities to atmospheric SO2 concentrations defines a composite geographical pattern across the considered basins"

5. Manuscript text: Conclusions. We found that, since 1850, high SO2 atmospheric concentrations are followed, during year +1, by significant delayed responses of both the North Atlantic SST and NAO index (Fig.5). This finding suggests a radiative forcing effect of sulphur rich eruptions, as we

Comment by referee#2: Figure 5 is quite confusing. It is not really well described in the next, and it is distracting that it appears cited in the discussion section, as I mentioned earlier. I have several concerns: why is the standard deviation of the SSTs calculated over the whole period shown ? It seems to me that the authors want to show a statistically significance difference in the mean of the SSts in post-volcanic years and in pristine years. The standard deviation is not informative because it is the difference in the mean of two populations. The significance of these differences depends on the magnitude of the difference, the pooled standard deviations and the sample size. This third factor is not included in the figure. The results concerning the NAO (lower panel) do not seem to indicate a strong, or even statistically significance systematic response. The red and black lines are pretty close to each other and most of the time within the uncertainty ranges (= standard deviations of the mean?) Again, I think that this figure s not showing how significant the response of the NAO is, and nevertheless, even the sign of the response changes with time lag. For instance, it seems significant for August and September and then not significant in June, Juy and October. Can this be just a random effect ?. What are the physical mechanism by which the sign of the response may change ?

Answer to comment: We agree. In the revised version of the manuscript figure 5 should be presented and discussed earlier. Actually, the standard deviation of the mean, ïĄşmean, as reported in our analysis (sea above definition), fully takes into ac-

count for both the magnitude of the differences of sea surface temperatures and the sample sizes (i.e., being the latter the number of years). We also note that the NAO values reported in the figure 5 are normalised, in order to filter possible multiyear trends in the NAO variations. In this regard, although there is an element of hazard in evidencing general conclusions from the NAO trend reported in figure 5, it makes sense that the effects of radiative forcing produced by SO2 (i.e., as evidenced by statistically significant differences between pristine year and SO2 polluted years) both on SST and NAO are higher during months with highest solar irradiation (i.e., as also evidenced by the referee#2 "it seems significant for August and September". Concerning the possible random effects of the observed trends, we are confident that the NAO standard deviation of the mean, as calculated from multi-decadal records, provides an evaluable threshold well beyond the NAO random variability.

6. Manuscript text: by climate models (Driscoll et al., 2012; Charlton-Perez et al., 2013). In this regard, the Atlantic sea surface temperature (SST) is one of the most important governing factors for the NAO and the atmosphere dynamics over most parts of the Northern Hemisphere (Hurrell, 1995). Moreover, the lagged decrease of the NAO index following SO2-induced

Comment by referee#2: This paragraph is to me quite problematic. The question of whether SST anomalies are driving NAO variability and atmospheric circulation at mid-latitudes in general is being discussed for at least 25 years, and there are studies with very opposing views. This section cites some of the studies by Czaja and Frankignoul and by Wang et al, that do indicate some response to SST anomalies, but there are may others that show no response. For instance, the paper by Sutton and Hodson (Science, 2005,doi:10.1126/science.1112666) indicates that ' However, thus far the evidence for an Atlantic link is mainly circumstantial, being derived from observations and showing correlation rather than causality' . This is a sign that, even in 2005, this question was far from settled, and actually it is not settled yet. Yet, the authors discuss this point rather superficially, as if it were widely accepted that NAO variability is driven

by North Atlantic SSTs. Even the studies showing a response of the NAO to SSTs admit that the signal is weak. The paper by Sutton and Hodson identifies a response in the low-frequency band of the spectrum, i.e. not at interannual time scales, which would be the relevant time scales here, but at decadal timescales. Furthermore, the paper by Hurrel et al (1995) does not mention that SSTs are the most important driver of the NAO. Actually, if I understood that paper properly, it does not deal with the driving factors of the NAO. The statistical connections between the NAO and SSTs shown there are interpreted as a response of the SSTs to NAO forcing, which is the opposite interpretation given by the authors here.

Answer to comment: We fully agree that, even in 2005, the question if whether SST anomalies are driving NAO variability and atmospheric circulation at midlatitudes. Also, we fully agree with referee#2 as our paper does not deal with the driving factors of the NAO. Our research was partially motivated by the results of a paper by Sottili (Constraints on climate forcing by sulphate aerosols from seasonal changes in Earth's spin; Geophys. Journ. Int, 2014 197, 1382–1386) concluding that, if we only consider the Earth's surface cooling induced by aerosol radiative forcing, we largely underestimate the effects of aerosols on atmospheric circulation. Specifically, the energy budget of atmospheric kinetic energy after large eruptions should be coherently explained only by assuming a strong influence of sulphate aerosols on partitioning the available energy into the atmosphere; for example, by assuming a strong inlfluence of sulphate aerosols on affecting the latent heat release and transport during condensation–evaporation–freezing cycles (Sottili, 2014). The futher steps must also take into account the widely reported decrease, both from historical records and climate modelling, in global mean precipitation by volcanic aerosols, as explained by the stabilization of the atmosphere due to the reduction of short-wave radiation reaching the surface. On these grounds, during the early stages of analysis, by considering the "classical" hydrological parametrs (i.e., mean annual and monthly precipitations and streamflow data, etc.) we found poorly or not significant evidences on volcanic forcing of the European hydrological cycle. After this initial and sterile stage of analysis, when the lack of a significant SO2 forcing seemed to contradict the hypotheses by Sottili (2014), we decided to explore an alternative path, as we investigated this issue from a kinetics point of view i.e., by considering the short-terms, daily, effects of SO2 on rainfall and streamflows intensities. This meant to face a much more difficult path for three main reasons: 1) the significant reduction of available data (daily records) from hydrological datasets; 2) no possibilities of comparison of our analysis on short term hydrological paramenters with available climate modelling outputs on rainfall and streamflow data; 3) the need of reconciling the clear contradiction between the reported global-scale "stabilization effects" of SO2, due to the reduction of short-wave radiation, and our results clearly showing an enhancement of short-term, extreme hydrological events both in terms of rainfall and streamflow intensities. On these grounds, we proposed a mechanism of interaction between SO2 and short-term extreme hydrological events involving a continental-scale redistribution of atmospheric available energy, with the SST (or, more precisely, with the amount of thermal energy stored in the Ocean) playing a key role in reconciling the contradiction between the reported long-term, global-scale "stabilization effects" of SO2 and our findings showing a clear enhancement of short-term extreme hydrological events. Although our interpretation about the role of SST and NAO needs further studies, hopefully from climate modelling and independent datasets, our contribution intends to stimulate the scientific debate on this issue.

7. Manuscript text: We propose a teleconnected mechanism for volcanically induced extreme hydrological events in Europe. Specifically, the triggering mechanism of extreme rainfall and streamflow events in Europe since 1850 after sulphur-rich eruptions can be explained by sulphate aerosol radiative forcing over North Atlantic causing a net decrease of heat exchange between Ocean and atmosphere through evaporation, precipitation and atmospheric- heating processes. The results of this study display how sulphur-rich eruptions have relevant significance in driving the frequency and intensity of rainfalland related floods in Europe, with variable effects in different climate zones.

Comment by referee#2: To be honest, I do not think that this study has shown anything

of this sort. It has not analysed radiative forcing, nor ocean-atmosphere heat exchange, nor evaporation, and it has analysed only four rain-gauges in Europe. I found the claims contained in this paragraph completely unsubstantiated.

Answer to comment: As discussed above, we fully agree with the referee#2 about the need of considering the radiative forcing as a key driving factor, which can also play a role in determining the observed pattern of extreme hydrological events. However, we again remark that the response of precipitation and run-off to volcanic sulphate aerosols eruptions does not indicate a stabilization of the atmosphere; rather, we observed a significant increase of extreme events strongly suggesting an increase of hydrological cycle dynamics (i.e., the opposite trend we should expect after the reduction of the radiative heating of atmosphere by sulphate aerosol forcing). In addition, we believe that short term changes of hydrological cycle dynamics (i.e., inter-annual higher extreme precipitations and related river discharge rates), are mechanism which are not yet well constrained by climate models, since they account for phenomena occurring at longer time scales (months rather than hours) and wider spatial scale (i.e., not considerning micro-climatic effects occurring at basin scale). On these grounds, we are confident that the triggering mechanism for extreme events by volcanic $SO_2$ is driven by rearrangement of available heat energy into the atmosphere (see for example Sottili, 2014).